# Identification of the NAC Transcription Factor Family during Early Seed Development in *Akebia trifoliata* (*Thunb*.) Koidz

**DOI:** 10.3390/plants12071518

**Published:** 2023-03-31

**Authors:** Huijuan Liu, Songshu Chen, Xiaomao Wu, Jinling Li, Cunbin Xu, Mingjin Huang, Hualei Wang, Hongchang Liu, Zhi Zhao

**Affiliations:** 1College of Life Sciences, Guizhou University, Guiyang 550025, China; 2Guizhou Key Laboratory of Propagation and Cultivation on Medicinal Plants, Guizhou University, Guiyang 550025, China

**Keywords:** NAC family, transcription factor, metabolites, early seed development, *Akebia trifoliata*

## Abstract

This study aimed to gain an understanding of the possible function of NACs by examining their physicochemical properties, structure, chromosomal location, and expression. Being a family of plant-specific transcription factors, NAC (petunia no apical meristem and *Arabidopsis thaliana* ATAF1, ATAF2, and CUC2) is involved in plant growth and development. None of the NAC genes has been reported in *Akebia trifoliata* (*Thunb*.) Koidz (*A. trifoliata*). In this study, we identified 101 NAC proteins (AktNACs) in the *A. trifoliata* genome by bioinformatic analysis. One hundred one AktNACs were classified into the following twelve categories based on the phylogenetic analysis of NAC protein: NAC-a, NAC-b, NAC-c, NAC-d, NAC-e, NAC-f, NAC-g, NAC-h, NAC-i, NAC-j, NAC-k, and NAC-l. The accuracy of the clustering results was demonstrated based on the gene structure and conserved motif analysis of AktNACs. In addition, we identified 44 pairs of duplication genes, confirming the importance of purifying selection in the evolution of AktNACs. The morphology and microstructure of early *A. trifoliata* seed development showed that it mainly underwent rapid cell division, seed enlargement, embryo formation and endosperm development. We constructed AktNACs co-expression network and metabolite correlation network based on transcriptomic and metabolomic data of *A. trifoliata* seeds. The results of the co-expression network showed that 25 AtNAC genes were co-expressed with 233 transcription factors. Metabolite correlation analysis showed that 23 AktNACs were highly correlated with 28 upregulated metabolites. Additionally, 25 AktNACs and 235 transcription factors formed co-expression networks with 141 metabolites, based on correlation analysis involving AktNACs, transcription factors, and metabolites. Notably, *AktNAC095* participates in the synthesis of 35 distinct metabolites. Eight of these metabolites, strongly correlated with *AktNAC095*, were upregulated during early seed development. These studies may provide insight into the evolution, possible function, and expression of AktNACs genes.

## 1. Introduction

The NAC petunia no apical meristem (NAM) and *Arabidopsis thaliana* ATAF1, ATAF2, and CUC2 transcription factors are one of the largest families found in plants [1,2]. They exhibit a highly conserved NAC subdomain at the N-terminal region consisting of approximately 150 amino acid residues, which contain five conserved regions (A–E) [3,4]. NAC proteins are homo- and hetero-dimerized, and the ability to dimerize is localized to the NAC domain [4,5]. They can form functional NAC protein dimers through salt bridging, with rich positive charges on one side of the surface, which may be related to DNA binding [6,7]. The C-terminal region is a highly differentiated and transcriptional regulatory regulator, characterized by the frequent occurrence of simple amino acid repeats and serine and threonine, proline and glutamine or acidic residues [8,9,10,11].

NAC transcription factors have a wide range of biological functions and play a vital role in plant growth and development, metabolic regulation, hormone response, stress resistance and crop quality improvement [12,13,14]. The first NAC transcription factor, NAM, was found to be associated with embryo growth and development in *Petunia* [1]. *Petunia* embryos carrying the no apical meristem (NAM) mutation failed to develop apical meristems, resulting in seedlings lacking roots and leaves. Both *GmNAC11* and *GmNAC20* are induced by biotic stress and phytohormones in soybean. Overexpression of the *GmNAC20* not only enhanced the tolerance of plants to salt and low temperature but also promoted the formation of lateral roots [15]. *SlNAM1*, a salt-sensitive NAC gene in tomato (*Solanum lycopersicum*), is highly expressed in flowers and ripe fruit. Furthermore, *SlNAC1* and *SlNAM1* in *tomato* cultivars can be induced by salt stress, suggesting a possible involvement in tomato resistance responses to adversity stress [16]. Notably, NAC genes are capable of regulating aspects of plant embryo development, seed size and seed development. The *BcNAC2* in turnips (*Brassica campestris*) was most abundantly expressed in young horn fruits at 8 days after pollination [17]. In situ hybridization in tissues showed that *BcNAC2* was highly expressed in the embryo sac, indicating that the NAC transcription factor is associated with seed or embryo development. *NARS1* and *NARS2* (also known as *NAC2* and *NAM*) in *Arabidopsis thaliana* control embryogenesis by regulating the growth and development and degeneration of the integument at the torpedo-shaped embryo stage [18]. *NARS1* and *NARS2* are not expressed at the torpedo-shaped embryo stage, but they are expressed in the outer integument. The double knockout mutant nars1/nars2 exhibits an abnormal seed shape and delayed degeneration of the integument. Pollination of wild-type plants with the double mutant nars1/nars2 produced normal seeds and, conversely, abnormal seeds. In rice, *ONAC020*, *ONAC026* and *ONAC023* showed a particularly high horizontal expression during seed development [19]. Knockout of watermelon *CLNAC68* using the CRISPR-CAS9 system leads to delayed seed development and germination inhibition in a watermelon mutant [14].

After double fertilization is completed in plants, the ovary begins to expand, accompanied by seed development. Seed development is a highly complex process involving changes in seed morphology, size, and metabolites, and it is regulated by numerous key genes. Metabolites during seed development are divided into primary and secondary metabolites. Primary metabolites include sugars, lipids, etc., which are important indicators of seed quality and provide essential nutrients and energy for seed development. Secondary metabolism is a normal physiological activity that has evolved over time and interacts with both biotic and abiotic environments, playing a crucial role in plant growth and development. NAC transcription factors are involved not only in regulating plant life processes such as seed embryo development, leaf senescence, flower formation, and root development but also in regulating primary and secondary metabolic processes. For instance, plants overexpressing *OsNAC23* in rice exhibited higher photosynthetic rates, sugar translocation, and larger organ sizes, resulting in sustained increases in rice yield [20]. *StNAC103* gene silencing in potato epidermis was associated with increased wax and wax loading, particularly in alkanes, ω-hydroxy acids, diacids, ferulic acid, and primary alcohols [21]. Transcript levels of several genes associated with flavonoid biosynthesis and anthocyanin levels were significantly higher in *Arabidopsis* plants overexpressing *ANAC078* (Ox-*ANAC078*) and significantly lower in knockout *ANAC078* plants (KO-*ANAC078*) [22]. Additionally, NAC transcription factors can regulate key genes in lignin synthesis, affecting lignin synthesis and cell wall structure. The *Eucalyptus* NAC transcription factor *EgNAC141* positively regulates lignin biosynthesis and increases lignin deposition [23].

At present, more data on NAC transcription factors of plants have been collected from studies based on plant genomics [24,25] and bioinformatics [26]. Moreover, NAC transcription factors have promising applications in plant molecular breeding. *Akebia trifoliata* is a woody vine plant belonging to genus *Akebia* (Lardizabalaceae), geographically distributed in eastern Asia (China, Japan, and Korea) [27], which is used as a traditional Chinese herb consisting of many bioactive compounds such as triterpenoid saponins and flavonoid [28]. The dried fruit alias “Akebiae Fructus”, comprises seeds, peel and pulp, which have anti-inflammatory, antibacterial, antioxidant and anticancer properties [29,30,31]. Interestingly, *A. trifoliata* can also be eaten as a fruit, as it has well-developed placental tissue and a sweet taste. It is rich in nutrients, including soluble sugars, vitamin C and many amino acids. However, its seed content holds 70% of the entire edible part of the fruit, affecting its delicacy. Furthermore, the severely impacted natural environment has led to a decline in wild resources, and there is an urgent need for artificial cultivation for the exploitation and conservation of resources.

The research team in this study completed the sequencing of the whole genome of *A. trifoliata* and obtained the first information on the NAC gene family. In this study, we aim to gain an understanding of the possible function of NACs by analysing physicochemical properties, structure, chromosomal location and expression. The results will provide a theoretical basis for genetic breeding and improvement work on *A. trifoliata* fruits.

## 2. Results

### 2.1. Identification and Analysis of NAC Genes in A. trifoliata

Based on the study of protein homology, 101 NAC genes were identified: *AktNAC001–AktNAC101* (Appendix A). The AktNAC gene family encodes proteins with the length ranging from 96aa (*AktNAC021*) to 691aa (*AktNAC024*), with predicted molecular weights of 11,463.13 (*AktNAC021*) to 77,893.7 Da (*AktNAC076*). Most of the NAC proteins (73.2%) had an isoelectric point (pI) < 7 and were acidic in nature, suggesting that AktNAC proteins were rich in acidic amino acids. The hydrophobicity index was <0, indicating that all NAC proteins were hydrophilic. The instability index ranged from 20.76 (*AktNAC013*) to 64.37 (*AktNAC063*), while the aliphatic index ranged from 51.79 (*AktNAC049*) to 83.36 (*AktNAC014*). The subcellular localization of proteins suggested that AktNACs were distributed in the cytoplasm, nucleus, plasma membrane, and chloroplast. Moreover, the number of potential phosphorylation sites ranged from 17 (*AktNAC016*) to 93 (*AktNAC036*).

We aligned 428 proteins to construct a phylogenetic tree to further examine the evolutionary relationship between NAC proteins of *A. trifoliata*, grape, rice, and *Arabidopsis thaliana* (Figure 1). The NAC proteins were classified into eight subfamilies (NAC-a, NAC-b, NAC-c, NAC-d, NAC-e, NAC-f, NAC-g, and NAC-h, NAC-i, NAC-j, NAC-k) via the phylogenetic analysis. The NAC-j and NAC-e groups contain the most AktNACs, 17 and 15, respectively, followed by the NAC-g (12), NAC-a (10), NAC-b (10) and NAC-d (10) groups, with NAC-l constituting the least number of AktNAC (1). The close phylogenetic relationship between *A. trifoliata* and *Arabidopsis thaliana* NAC proteins suggested that they might have similar structures or biological functions. For example, *ANAC072* showed a significantly increased tolerance to drought [32] and the novel abscisic acid (ABA)–dependent stress-signaling pathway [33], indicating that *AktNAC053* and *AktNAC084* might be involved in drought tolerance and in ABA response.

We performed a multiple sequence alignment and phylogenetic analysis of the protein sequences from the NAC gene family in *A. trifoliata*, constructing an evolutionary tree (Appendix A). The tree classified the AktNAC gene family into 11 subclasses, including OsNAC8, TIP, NAC2, ANACO11, NAM, OsNAC7, OsNAC022, ATAF/NAP, ONAC079, ANAC001, and ONAC003/ANAC063. Furthermore, we conducted a conserved motif analysis of the AktNAC protein sequences. The analysis revealed that the majority of AktNAC proteins contain subdomains A–E at the N-terminus. Motifs 3, 4, and 1 were designated as NAM sub-domains A, B, and C, respectively. Additionally, motif 6 and motif 2 were identified as subdomains D and E, respectively. AktNAC proteins within the same subclasses exhibit broadly similar structures. For instance, OsNAC8 and TIP, belonging to the NAC-e group, have the highest number of motifs. NAM and OsNAC7 belong to the NAC-a group and are similar in length, both containing six identical motifs. NAC2 and ANAC011 subclasses, belonging to the NAC-d group, possess a similar arrangement and position of motifs. ANAC063 subclass proteins contain motifs 1–6, and their structural diversity may be associated with their distinct functions.

We analyzed the gene structure of members within the NAC gene family, presenting information on exons and introns. This analysis aids in understanding the structural conservation and variability of the gene family during evolution. Introns contribute to proteome complexity by producing different combinations of exons, which in turn translate into various proteins through alternative splicing [34,35]. The arrangement of exons and introns offers valuable insights into the evolution of gene families [36]. With the exception of the NAC-a, NAC-e, and NAC-d subclasses, other AktNAC subclasses possess three exons, a pattern also observed in *Arabidopsis thaliana* [4]. Additionally, 33 genes contained 5′-UTRs, and 31 genes contained 3′-UTRs. We further analyzed the conserved structural domains of selected AktNACs (Figure 2), demonstrating that substructural domains A, C, and D of AktNACs are highly conserved. This high conservation suggests that domains A, C, and D play crucial roles in the function of NAC family genes, a situation similar to that in *Arabidopsis thaliana* [37]. Previous research has reported that subdomain C may be involved in DNA binding [38]. The sequence diversity in subdomains B and E could imply distinct roles for these sequences within the NAC domain.

### 2.2. Chromosomal Localization and Duplication of NAC Genes in A. trifoliata

We identified the chromosomal location of 101 AktNACs (Figure 3), scattered unevenly across 16 chromosomes. Chr01 constituted the most AktNAC genes (twenty), while chromosome 12 comprised only one.

Gene duplication events have a significant effect on the gene family abundance and genome complexity in Eukaryotes [39]. We analyzed the duplication events of this family to understand the possible relationships between the potential gene duplications of AktNACs (Appendix A). Gene families form as species diverge and their evolution can be assessed using PAL2NAL to calculate synonymous (Ks) and nonsynonymous (Ka) values, thus revealing the history of gene evolution [40]. Therefore, Ks and Ka values determined whether the selection acted on protein-coding genes [41]. We estimated Ka/Ks values for 44 pairs of duplicated genes (Appendix A). We identified 13 tandem replication gene pairs and 31 segmental ones. The Ka/Ks ratio was <1 suggested that purifying selection acted on these duplicated gene pairs [42].

To further investigate the phylogenetic mechanism of the NAC gene family in *A. trifoliata*, a collinearity analysis was performed between *A. trifoliata* (A), *Arabidopsis thaliana* and the published genome of the *A. trifoliata* (B) (Appendix A). The number of co-linear gene pairs between *A. trifoliata* (A) and *Arabidopsis thaliana* was 2. The number of co-linear gene pairs between *A. trifoliata* (A) and *A. trifoliata* (B) was 76, distributed on 15 chromosomes, with no co-linear gene pairs present on chromosome 2. This indicates that the *A. trifoliata* genome (A) is closely related to the published *A. trifoliata* genome(B) and has a high degree of NAC gene homology.

### 2.3. Analysis of Promoter Cis-Acting Elements and Protein Interaction Networks

Gene function is found to be linked to differences in promoter regions [43]. This difference is primarily due to the type and number of cis-acting elements involved in the regulation of gene expression. Hence, we identified 52 cis-acting elements associated with hormone response, light responsiveness, stress responsiveness, and tissue-specific expression, distributed in the 2000-bp upstream region of 101 AktNAC genes (Appendix A). In terms of hormone response, ABA-responsive elements (ABREs), CGTCA motif (MEJA response), TCA-element (salicylic acid response), and TGACG motif (MEJA response) were present in most AktNAC genes, with *AktNAC084* comprising 25 ABREs. The light-responsive elements have the largest number of types (31), of which six elements, Box 4, GATA motif, G-box, GT1 motif, I-box, and TCT motif, are present in most AktNAC genes. Elements such as 3-AF1 binding site, AAAC motif, AT1 motif, CAG motif, Gap box, ACA-motif and Sp1 are present in fewer genes, while GATA-motif, chs-Unit 1 m1, and ACA-motif were present only in *AktNAC057*, *AktNAC064*, *AktNAC092*. Tissue-specific expression-related elements, such as CAT box, GCN4 motif, MBSI, O2 site, and RY element, were also identified, which were associated with phloem expression, endosperm expression, flavonoid biosynthesis, and nitrogen-responsive and seed-specific expression, respectively.

### 2.4. Transcriptomic Analysis of the Early Developmental Process in the Seeds of A. trifoliata

Seed development can have three stages: early embryogenesis, maturity, and post-embryogenesis [44]. During the early-embryogenesis stage, seed weight and lipid content are low. At the maturity stage, the seeds increase rapidly in terms of dry weight and accumulate large amounts of storage oil and storage protein. In the post-embryogenesis stage, the dry weight of the seeds remains constant while severe water loss occurs. The seed structure of *A. trifoliata* showed that 0DAF–70DAF (days after flower) is at the early stage of embryogenesis when seed endosperm cells divide rapidly and actively and the seeds increase rapidly in size (Figure 4). Moreover, we analysed the expression of AktNACs genes during the early stages of seed development using transcriptomic data. Based on FPKM > 1, we plotted a heatmap of AktNACs (Figure 5). Among them, 63 genes had expression, and 38 genes had no detectable expression (FPKM < 1). We filtered differential genes based on the value of |log2FC > 1| and FDR < 0.01. Among the genes analyzed, 63 exhibited expression, while 38 had no detectable expression (FPKM < 1). We filtered for differentially expressed genes using the criteria |log2FC > 1| and FDR < 0.01. Comparing 30 days after flowering (DAF) versus 70 DAF, we identified 30 differentially expressed AktNAC genes, with 10 genes (*AktNAC022*, *AktNAC021*, *AktNAC60*, *AktNAC85*, *AktNAC95*, *AktNAC064*, *AktNAC034*, *AkNAC047*, *AktNAC062*, *AktNAC071*) showing significant upregulation during seed development, and 17 AktNACs (*AktNAC030*, *AktNAC053*, *AktNAC035*, *AktNAC031*, *AktNAC036*, *AktNAC067*, *AktNAC032*, *AktNAC037*, *AktNAC098*, *AktNAC072*, *AktNAC059*, *AktNAC088*, *AktNAC033*, *AktNAC049*, *AktNAC007*, *AktNAC011*, *AktNAC058*) exhibiting significant downregulation. These results suggest that NAC genes play an important role in early seed development. We analyzed the expression of seven AktNAC genes during the early seed development stage using quantitative real-time polymerase chain reaction (qRT-PCR; Appendix A). Gene names and primer sequences are presented in Appendix A.

### 2.5. Co-Expression Network and Enrichment Analysis of Differential Genes during Early Seed Development of A. trifoliata

NAC genes are usually involved in the regulation of plant seed growth and development together with other transcription factors. To explore the regulatory network between AktNACs genes and other transcription factors during early seed development, we performed transcription factor identification of transcriptome data during early seed development. A total of 1647 transcription factors were identified and classified into 58 types. The first five families of transcription factors with the highest numbers were bHLH (155); MYB (126); NAC (109); ERF (102); and C2H2 (102). Based on Pearson’s correlation coefficient ≥ 0.9, a co-expression network analysis was performed using a Python script to explore AktNACs co-expression transcription factors, and the results revealed (Figure 6) that the network had 258 nodes and 606 network pairs, with a total of 25 AktNAC genes co-expressed with 233 transcription factors (*r* > 0.9 or *r* < −0.9), with the top five transcription factors in highest number being bHLH (19); GRAS (14); HD-ZIP (13); ARF (12); ERF (12); MYB (12).

In parallel, we annotated and performed pathway analysis of 25 AktNACs and 233 transcription factors in the co-expression network (Appendix A). Seed development undergoes cell division and differentiation and presents in structural and morphological changes, with the embryo maintaining mitotic activity through the continuous accumulation of starch converted to glucose. Sucrose upregulates the expression of genes associated with storage proteins and initiates the cellular differentiation process [44]. Plant hormones are also involved in the regulation of seed development. During seed development genes related to transport and energy are highly transcribed and energy metabolism is constantly changing. These genes are annotated in biological processes mainly related to the cellular process, metabolic process, biological regulation, and development process [44]. In biological process, genes were involved in positive regulation of transcription, DNA-templated, positive regulation of RNA biosynthetic process, positive regulation of nucleic acid-templated transcription, response to gibberellin, regulation of flavonoid biosynthetic process, floral organ development, etc. In cellular component category, genes were involved in transcription factor complex, endoplasmic reticulum membrane, nuclear outer membrane-endoplasmic reticulum membrane network, endoplasmic reticulum subcompartment, endoplasmic reticulum part and others. In molecular function, genes were involved in regulatory region nucleic acid binding, transcription regulatory region DNA binding, protein dimerization activity, transcription factor binding and others.

We further performed the GO and KEGG enrichment analyses (Appendix A). GO enrichment analysis indicated that genes were involved in co-expressed network regulatory region nucleic acid binding (53), transcription regulatory region DNA binding (53), positive regulation of transcription, DNA-templated (52), positive regulation of RNA biosynthetic process (53), positive regulation of nucleic acid-templated transcription (53) and others. KEGG enrichment analysis indicated that genes were involved in plant hormone signal transduction (22), circadian rhythm—plant (4), MAPK signaling pathway (5).

### 2.6. AkNACs Involved in Metabolic Regulation in Early Seed Development

By integrating the transcriptome and metabolome data, we performed an association analysis of the differential NAC and all upregulated metabolites in the transcriptome found in the NAC and metabolite association analysis (Figure 7). There were 51 nodes with 175 network pairs (*r* > 0.95) in the co-expression network, including 23 AktNACs and 28 metabolites, of which 122 network pairs are negatively correlated and 53 network pairs are positively correlated. 7 genes with Degree ≥ 10 were *AktNAC050* (17), *AktNAC095* (16), *AktNAC058* (13), *AktNAC089* (11), *AktNAC035* (10), *AktNAC031* (10), *AktNAC060* (10). Six metabolites with Degree > 10 were Procyanidin B1(15), L-Methionine (14), N6-Succinyl Adenosine (14), Cytidine (14). Methylmalonic acid (13), Kaempferol 7-*O*-glucosdie (12). *AktNAC095* is a significantly up-regulated gene in seed development in Trifolium. By correlation analysis with metabolites (Table 1), we found that eight metabolites strongly associated with *AktNAC095*, Kinic acid, Trifolin, Histamine, γ-Aminobutyric, Cytosine, Uridine 5′-diphospho-d-glucose, Kaempferol 7-O-glucosdie, d-Glucose 6-phosphate expression was significantly up-regulated during seed development (Figure 8).

### 2.7. Co-Expression Analysis of AktNAC, TFs and Metabolites

We performed co-expression network analysis of AktNACs, transcription factor and relative peak areas of metabolites (Figure 9). The results showed that there were 401 nodes in the co-expression network, containing 25 AktNACs, 235 transcription factors and 141 metabolites. The positive correlation network was 596 pairs (*r* > 0.9), and the negative correlation network was 263 pairs (*r* < −0.9). The degree of correlation between each node was expressed as Pearson’s correlation coefficient (r), where the top 5 AktNACs were *AktNAC030* (133), *AktNAC032* (106), *AktNAC089* (83), *AktNAC037* (79), *AktNAC093* (65), with degrees between 10 and 60 containing 12 AktNACs. In the co-expression process, there were multiple transcription factors associated with AktNACs, and the transcription factors with degree > 5 contained 14 transcription factors, namely *mikado.Chr08G960* (TALE; degree = 7), *mikado.Chr06G1013* (BES1, degree = 7), *mikado.Chr03G263* (ERF, degree = 6), *mikado.Chr04G2651* (HD-ZIP, degree = 6), *mikado.Chr13G492* (HD-ZIP, degree = 6), *mikado.Chr12G1156* (bHLH, degree = 6), *mikado.Chr11G1290* (GRAS, degree = 6), *mikado.Chr09G896* (MIKC_MADS, degree = 6), *mikado.Chr07G2034* (CO-like, degree = 6), *mikado.Chr06G1656* (CO-like, degree = 6), *mikado.Chr06G1417* (HD-ZIP, degree = 6), *mikado.Chr06G1350* (GRAS, degree = 6), *mikado.Chr02G2324* (WRKY, degree = 6), *mikado.Chr01G2513* (LBD, degree = 6).

It is hypothesized that AktNACs synergistically regulated the biosynthesis of metabolites with transcription factors, the most typical of which are *AktNAC030* and *AktNAC32* through synergy with multiple transcription factors, with *AktNAC030* co-expressing with 96 transcription factors and *AktNAC32* co-expressing with 63 transcription factors, resulting in multiple transcription factors synergistically regulating NAC transcription factors. Among the transcription factors, *AktNAC032* and *AktNAC095* were the most typical ones, with *AktNAC032* regulating the synthesis of 41 metabolites (*r* > |0.9|) and *AktNAC095* regulating the synthesis of 35 metabolites (*r* > |0.9|). Integration analysis of AktNACs, transcription factors and metabolites, we identified the most likely transcription factors that synergistically regulate metabolites with *AktNAC060*, *AktNAC095*. This indicates that NAC plays an important role in the synthesis of compounds and that transcription factors are co-expressed in concert with NAC in order to achieve regulation of metabolite biosynthesis (Table 2).

### 2.8. Protein Interaction Networks of AktNAC

Protein interaction networks consist of interacting proteins that participate in various aspects of biological processes, such as signaling and regulation of gene expression. Systematic analysis of a large number of protein interactions is crucial for understanding the functional connections between proteins. To further comprehend the functions of 25 AktNACs within the co-expression network of AktNACs, transcription factors (TFs), and associated metabolites, we constructed a NAC protein interaction network based on Arabidopsis homologs (Figure 10). Among these, 12 genes from M. tridentata were involved in the network. Five AktNACs were directly linked to NTL (homolog of *AktNAC038*), and NTL was linked to SOG1 (homolog of *AktNAC058*), each acting within their respective clusters. ATAF1 (homolog of *AktNAC006*) serves as the nodal gene connecting the two clusters. Gene Ontology (GO) predictions of gene functions within the network include nucleus, DNA binding, organic cyclic compound binding, regulation of cellular metabolic processes, regulation of macromolecule metabolic processes, regulation of transcription (DNA-templated), regulation of nucleobase-containing compound metabolism, cell cycle, and DNA repair (Appendix A).

## 3. Discussion

In this study, we identified, for the first time, members of the NAC gene family at the genomic level using the transcriptome and metabonomics to characterize expression during the early seed development stage, which provided a valuable resource for further studies on the function of *A. trifoliata* NAC genes.

We identified 101 AktNAC genes encoding these protein families, a number close to *Arabidopsis thaliana* (105) [37] and pepper (104) [45], but less than rice (151) [9] and maize (152) [46], and more than grape (74) [47] and mung bean (81) [48]. These results suggest that the number of NAC gene families varies across species, which may be due to the evolutionary expansion of NAC gene families. Exons and introns are essential features of gene evolution. The AktNAC gene exons identified in this study ranged from one to nine, with most AktNAC genes having three exons, similar to species such as walnut [49], cucumber [50], and *Chenopodium quinoa* [51]. This may be related to the functional differentiation and structural diversity of the AktNAC gene family. There are differences in the arrangement of conserved AktNAC motifs located in different groups, but most AktNAC genes have a similar motif composition, which corresponds to the NAC gene structure and function.

In this study, 101 AktNACs were classified into 11 subgroups based on the results of phylogenetic tree analysis of *A. trifoliata* and *Arabidopsis*, grape, and rice genes. NAC-a (NAM subclass) is mainly involved in embryo, stem tip, flower, and root growth and development [1,2], while NAC-g and NAC-k (ATAF/NAP subclass) usually regulate different types of stresses [52]. This suggests that AktNAC genes have multiple biological functions in plant growth and development and stress response.

Gene duplication can provide an evolutionary impetus for species to generate new functions [53]. Genome-wide duplication, which occurs in genetic lines of several domesticated crops including wheat (*Triticum aestivum*), cotton (*Gossypium hirsutum*), and soybean (*Glycine max*), contributes to significant agronomic traits, such as grain quality, fruit shape, and flowering time [39]. Segmental duplications and tandem duplications are the main types of eukaryotic duplication events that occur [54]. We identified 44 pairs of duplicated genes, with the earliest duplication occurring at 94.629 MYA (million years ago) and the most recent duplication occurring at 3.578 MYA (Appendix A). In this study, the Ka/Ks ratio was mostly <1, indicating that the duplicated AktNAC genes were subjected to strong purifying selection pressure. The results suggested that segmental duplication contributed significantly to the expansion and evolution of the NAC gene family in *A. trifoliata*.

The morphology and microstructure of *A. trifoliata* seeds showed that before 70DAF the seeds mainly underwent cell division and differentiation, a process involving rapid increase in volume, with the endosperm gradually forming to cover the entire seed and the embryo developing into a globular embryo. Approximately 70DAF–150DAF seeds showed a less pronounced increase in volume, and the embryo gradually developed into a rod-shaped structure. The transcriptome data revealed the expression pattern of *A. trifoliata* seeds during the early developmental stage. In this study, we found that six differentially expressed class I NAC genes were significantly upregulated at 70 DAF, including four in the NAM subclass (*AktNAC060*, *AktNAC064*, *AktNAC047*, *AktNAC071*), one in the OsNAC8 subclass (*AktNAC34*), and one in the ATAF/NAP subclass (*AktNAC085*). Additionally, there were four class II NAC genes (*AktNAC62*, *AktNAC95*, *AktNAC22*, *AktNAC21*) significantly upregulated at 70 DAF. Thirteen differentially expressed class I NAC genes showed significant downregulation, including seven in the OsNAC8 subclass and four in the ATAF/NAP subclass. Furthermore, four class II NAC genes were observed. In *Arabidopsis thaliana*, class I NACs exhibit greater conservation in subdomains B and E compared to class II NACs, a pattern also observed in *A. trifoliata*. The differential expression of class I and class II AktNAC genes in seeds may be related to their structural differences in the B and E subdomains. *AktNAC039*, *AktNAC040*, *AktNAC085*, and *AktNAC100* were homologous to *Arabidopsis thaliana NAC056*, which when combined with *NAC018*/*NARS2* regulated embryogenesis by expressing the development and degeneration of the ovule integument, a process related to intertissue communication between the embryo and the maternal ovule integument [18]. In this study, *AktNAC085* was upregulated, while embryo formation started after 50 DAF, indicating that *AktNAC085* is highly likely involved in the regulation of embryonic development. The interaction of embryonic development gene *VvNAC26* (*GSVIVT01019952001*), homologous to *AktNAC059* and *VvMADS9* affects the development of grape seeds and fruits [55]. Expression of the *VvNAC26* gene was more pronounced in the ovules of seedless grape varieties than in the ovules of seeded grape varieties. Moreover, overexpression of *VvNAC26* in tomato resulted in smaller fruits and seeds of transgenic tomato *VvNAC26*, and significantly fewer seeds at the fruit ripening stage than nontransgenic tomato and transgenic control containing only empty vector. In this study, *AktNAC059* expression was higher at 30 DAF than at 50 DAF and 70 DAF, indicating that *AktNAC059* may be expressed earlier in seed development to influence seed size and number.

Plant growth and development are regulated by the expression of a large number of genes, including many specifically expressed transcription factors. Studies have shown that NAC transcription factors are involved in fruit development and ripening, embryonic development, stress response, and phytohormone signaling. *FtNAC70* in buckwheat is highly expressed under exogenous PEG, NaCl, cold, MeJA, ABA and GA induction. Tartary Buckwheat plants overexpressing *FtNAC70* showed greater salt and drought tolerance resistance [56]. *Arabidopsis thaliana* ATAF2 promotes ethylene biosynthesis and response through the activation of related genes [57]. Plant secondary metabolites are nonessential small molecule compounds for the proper functioning of plant growth and development, which are usually not directly involved in growth and development and have an important role in plant ecological adaptation [58]. The NAC transcription factors can synergistically regulate the biosynthesis of flavonoids, caffeine and theanine with MYB, bHLH and other transcription factors [59]. *CsNAC7* positively regulates the caffeine synthase gene *yhNAMT1*, whose overexpression increases caffeine accumulation in transgenic tea healing tissues [60]. *Watermelon ClNAC68* inhibits the expression of IAA-amino acid synthase (ClNAC3.6) in the IAA signaling pathway, positively regulates IAA accumulation in fruit and promotes normal seed development [14]. Overexpression of Norway spruce *PaNAC03* leads to downregulation of CHS, F3′H and PaLAR3 expression in the flavonoid synthesis pathway, resulting in abnormal embryo development [61]. In this study, transcriptome and metabolome association analysis of seed development in *A. trifoliata* identified strong correlations (*r* > 0.95) between AktNACs and many metabolites, which mainly include organic acids, alkaloids, flavonoids, lignans, coumarins, terpenoid and lipids. There are also numerous transcription factor families that are synergistically regulated with AktNACs, including MYB, bHLH, WRKY, C3H, B3, AP2 and others. This indicates that AktNACs have an important role in the synthesis of metabolites during the development of *A. trifoliata* seeds, which is consistent with literature reports. However, further studies are needed for their specific regulatory relationships and functions.

Protein interactions have also demonstrated that five AktNACs (*AktNAC042*, *AktNAC071*, *AktNAC089*, *AktNAC050*, and *AktNAC038*) are linked to NTL proteins, which are NAC with TM motif1-like (NTL) proteins, similar to tail-anchored (TA) protein structures. *NTL11* has been previously shown to upregulate the expression of genes associated with flavonoid biosynthesis, leading to the accumulation of anthocyanins in response to intense light stress [22]. *NTL9* is a negative regulator of formative layer activity and plays a repressive role in the developmental transition of the stem vascular system to the secondary growth stage [62]. Consequently, *AktNAC042* and *AktNAC038* may be involved in the regulation of flavonoid biosynthesis and secondary vascular development. SOG1 is thought to be implicated in the DNA damage response of plant p53 [63]. Moreover, other related NAC factors, such as *NAC044*, have also been shown to be involved in the DNA damage response [64]. This suggests that their homologs, *AktNAC058* and *AktNAC011*, may also mediate the DNA damage response in *A. trifoliata*. *NAC083*, also known as *VNI2*, exhibits specificity in regulating xylem cells [65]. *RD26* regulates the ABA and the brassinosteroid signaling pathway [33,66]. Therefore, *AktNAC095* and *AktNAC084* may mediate xylem differentiation and regulation of ABA signaling, while *AktNAC071* (a homolog of *Arabidopsis NAC100*) may bind to the promoter regions of genes involved in chlorophyll catabolic processes. In summary, *AktNAC071* and *AktNAC095* are important candidates for photosynthesis, flavonoid biosynthesis, and stress response in *A. trifoliata* seed development, and their functions require further validation.

## 4. Materials and Methods

### 4.1. Plant Material

In this study, wild germplasm was transplanted in the teaching farm of Guizhou university 3 years ago, whose plant origin is *Akebia trifoliata* (*Thunb*.) Koidz, belonging to Lardizabalaceae, genus *Akebia* Decne., identified by the plant identification center of Guizhou University. In this study, different stages in the same tree were harvested and then classified according to the seed development stage, with different developmental stages used as different samples. The tree was labeled on the day it fully blossomed (20 April 2019). Different fruits were randomly collected every 20 days during early seed development (20 May, 10 June and 1 July 2019), including 30 DAF (30 days after flower), 50 DAF (50 days after flower), 70 DAF (70 days after flower). Pulp and peel were quickly removed, and a total of 3 biological replicates were collected in each stage. The tender leaves were also collected from the tree. Samples were frozen in liquid nitrogen and stored at −80 °C with leaves for genomic analysis, and seeds for transcriptome and metabolome analysis.

### 4.2. Total RNA Extraction, cDNA Library Construction, and RNA-Seq

Total RNA was extracted from A. trifoliata seeds using the Trizol kit (Invitrogen, CA, USA). RNA degradation was assessed using an Agilent 2100 Bioanalyzer (Agilent Technologies, CA, USA) and RNase-free agarose gel electrophoresis. Total RNA was removed from rRNA and reverse transcribed into cDNA using random primers. cDNA libraries were constructed and sequenced using Illumina HiSeqTM 4000 (Gene Denovo Biotechnology Co., Guangzhou, China).

Raw data were run through fastp [67] for quality control to remove reads containing adapters, those containing >10% N, and those of low quality (number of bases with a quality value of Q ≤ 20 accounting for more than 50% of the entire read). The filtered data were mapped to the *A. trifoliata* reference genome using HISAT2 software (version 2.1.0) [68]. Transcripts were reconstructed using StringTie (version 1.3.4) [69], and the expression of genes in each sample was calculated with RSEM using the fragments per kilobase per million mapped reads (FPKM) method. We calculated the differentially expressed genes between the three different samples according to the FPKM method. A model with a negative binomial distribution was used for the three samples using DESeq2 [70]. The *p*-values obtained by DESeq were corrected using the Benjamini–Hochberg program, and the corrected *p*-value was used to detect the false discovery rate (FDR). We identified the differential genes by log2Fold Change <1 & >1, and FDR < 0.01.

### 4.3. Identification of NAC Genes in A. trifoliata

The genome, coding sequence, protein sequence, and Gff annotation file of *A. trifoliata* were assembled by sequencing at Genedenovo Biotechnology Co., Ltd. (Guangzhou, China). The *Arabidopsis* NAC protein sequence was downloaded from the Arabidopsis Information Resource (TAIR, https://www.arabidopsis.org/ accessed on 18 April 2022), and the grape and rice were downloaded from plantTFDB (http://planttfdb.gao-lab.org/index.php accessed on 18 April 2022). The Arabidopsis NAC protein was used as the query sequence, and the predicted protein sequence of A. trifoliata genome was used as the database for identification using the BLAST alignment method with TBtools (v. 1.0987663) [71]. The obtained AktNAC genes were identified using Pfam (http://pfam.xfam.org/family accessed on 18 April 2022), NCBI-CDD (https://www.ncbi.nlm.nih.gov/cdd/ accessed on 18 April 2022), and SMART (http://smart.embl-heidelberg.de/ accessed on 18 April 2022) online tools for NAC protein conserved structural domain prediction, excluding false-positive genes.

### 4.4. Basic Properties and Phylogenetic Analysis of AktNACs

The basic properties of NAC family proteins were analyzed using ExPaSy’s online Protparam software (https://web.expasy.org/protparam/ accessed on 22 April 2022). The hypotheses related to subcellular localization were analyzed using PSORTb version 2.0 [72] (https://www.psort.org/psortb/index.html accessed on 22 April 2022). The phosphorylation sites of proteins were predicted using NetPhos (v. 3.1) (https://services.healthtech.dtu.dk/service.php?NetPhos-3.1 accessed on 22 April 2022). The Muscle in MEGA 7.0 [73] was used to perform multiple sequence comparisons of the identified white mullein and Arabidopsis NAC proteins. We adopted the neighbor-joining method to construct the phylogenetic tree, with parameters set to the Poisson correction, P-distance, and bootstrap value of 1000 replicates, and other parameters by default. The phylogenetic trees were displayed using the online software iTOL (v. 6) [74] (https://itol.embl.de/ accessed on 22 April 2022).

### 4.5. Gene Structure, Conserved Motif, and Cis-Element Analyses of AktNACs

Based on the information from the Gff annotation file of A. trifoliata, the gene structure was analyzed using TBtools (v. 1.0987663) to characterize and map the gene structure. Conserved sequences of AktNACs were analyzed using the MEME version 5.0.5 [75] (http://meme-suite.org/tools/meme accessed on 26 April 2022), with the number of searches set to 10 for motif and other parameters by default plotted using TBtools (v. 1.0987663).

The 2000-bp sequence upstream of the transcriptional start site of AktNACs was extracted using TBtools (v. 1.0987663), and then submitted to PlantCARE (http://bioinformatics.psb.ugent.be/webtools/plantcare/html/ accessed on 26 April 2022) [76] to retrieve and analyze the cis-acting elements of the promoter region. The protein interaction network map was constructed using the STRING database (https://cn.string-db.org/ accessed on 26 April 2022) [77]. Moreover, the GO enrichment analysis was performed on the protein interaction network map node genes and plotted using the R package.

### 4.6. Chromosomal Location and Gene Duplication

The distribution location of the AktNAC on the chromosome was determined from the gene annotation data and was drafted using the R package. Duplication events of genes were analyzed using the Multiple Collinearity Scan toolkit (MCScanX) [78] with the default settings. The differentiation rate of replicated genes was calculated using the Ka/Ks ratio of the nonsynonymous substitution rate (Ka) to the synonymous substitution rate (Ks). Genes with Ks >2.0 were discarded to avoid substitution saturation. The divergence time was calculated using the following formula: T (T = Ks/(2r) × 10^−8^), where r is 1.5 × 10^−8^ synonymous substitutions per site per year for dicotyledonous plants [79].

### 4.7. Paraffin-Embedded (FPE) Tissue Sections of A. trifoliata Seeds

Seeds at 30 DAF, 50 DAF, 70 DAF, and 130 DAF were taken, photographed, and recorded; their appendages were removed, washed in distilled water, and blotted on filter paper; seeds were cut with a scalpel; the taken material was placed in an anatomical flask with FAA fix solution, pumped for more than 30 min until the material sank to the bottom of the vessel, and fixed with FAA fix solution for at least 24 h. (1) Paraffin sectioning: the sections were sequentially placed in xylene I for 20 min, xylene II for 5 min, anhydrous ethanol II for 5 min, 75% alcohol for 5 min, and then washed using tap water. (2) Safranin O staining: slices were stained in safranin O staining solution for 1–2 h and slightly washed using tap water to remove excess dye. (3) Decolorization: each slice was placed in 50%, 70%, and 80% gradient alcohol for 3–8 s. (4) Fast green staining: slices were placed in a fast green staining solution for 30–60 s, stained for 30–60 s, and dehydrated using anhydrous ethanol. (5) Transparent sealing: slices were placed in fresh xylene for 5 min and subjected to the neutral resin sealing. (6) Finally, microscopic examination, image acquisition, and analysis were conducted.

### 4.8. Metabolomics Experimental and Analytical Methods

All chemicals and reagents were of analytical grade. Methyl alcohol, acetonitrile and ethyl alcohol were purchased from Merck Company, Germany (www.merckchemicals.com accessed on 21 April 2022). Milli-Q system (Millipore Corp., Bedford, MA, USA) ultrapure water was used throughout the study. Authentic standards were purchased from BioBioPha Co., Ltd. Kunming, China (www.biobiopha.com/ accessed on 21 April 2022) and Sigma-Aldrich, St. Louis, MO, USA (www.sigmaaldrich.com/unitedstates.html accessed on 21 April 2022).

The freeze-dried samples were crushed using a mixer mill (MM 400, Retsch) with a zirconia bead for 1.5 min at 30 Hz. Then 100 mg powder was weighed and extracted overnight at 4 °C with 1.0 mL 70% aqueous methanol containing 0.1 mg/L lidocaine for internal standard. Following centrifugation at 10,000× *g* for 10 min, the supernatant was absorbed and filtrated (SCAA-104, 0.22-μm pore size; ANPEL, Shanghai, China, www.anpel.com.cn/ accessed on 2 May 2022) before LC–MS/MS analysis.

The compounds extracted were analyzed using an LC-ESI-MS/MS system (UPLC, Shim-pack UFLC SHIMADZU CBM30A, http://www.shimadzu.com.cn/ accessed on 16 May 2022; MS/MS (Applied Biosystems 6500 QTRAP, http://www.appliedbiosystems.com.cn/ accessed on 16 May 2022) [80]. Two μL of samples were injected onto a Waters ACQUITY UPLC HSS T3 C18 column (2.1 mm × 100 mm, 1.8 µm) operating at 40 °C and a flow rate of 0.4 mL/min. The mobile phases used were acidified water (0.04% acetic acid) (Phase A) and acidified acetonitrile (0.04% acetic acid) (Phase B). Compounds were separated using the following gradient: 95:5 Phase A/Phase B at 0 min; 5:95 Phase A/Phase B at 11.0 min; 5:95 Phase A/Phase B at 12.0 min; 95:5 Phase A/Phase B at 12.1 min; 95:5 Phase A/Phase B at 15.0 min. The effluent was connected to an ESI-triple quadrupole-linear ion trap (Q TRAP)–MS. LIT and triple quadrupole (QQQ) scans were acquired on a triple quadrupole-linear ion trap mass spectrometer (Q TRAP), AB Sciex QTRAP6500 System, equipped with an ESI-Turbo Ion-Spray interface, operating in a positive ion mode and controlled by Analyst 1.6.1 software (AB Sciex). The operation parameters were as follows: ESI source temperature 500 °C; ion spray voltage (IS) 5500 V; curtain gas (CUR) 25 psi; the collision-activated dissociation (CAD) was set highest. QQQ scans were acquired as MRM experiments with optimized declustering potential (DP) and collision energy (CE) for each individual MRM transitions. The *m*/*z* range was set between 50 and 1000.

### 4.9. qRT-PCR Verification of AktNACs

Seven unigenes were randomly selected to verify the reliability of the transcriptome data, and primers were designed using the Primer 5.0 software. Total RNA was extracted using cetyl trimethylammonium bromide considering RG5 (EF-α) as an internal reference gene. Approximately 16 µL of the gDNA removal system was prepared on ice using the iScript™ gDNA Clear cDNA Synthesis kit, and then this was incubated at 25 °C for 5 min and at 75 °C for 5 min to remove the gDNA. Then, 20 µL of the reverse transcription system was prepared on ice, and incubated at 25 °C for 5 min, 46 °C for 20 min, and 95 °C for 5 min. Subsequently, 20 µL of the reverse transcription system was prepared on ice, incubated at 25 °C for 5 min and at 46 °C for 20 min, denatured at 95 °C for 1 min, cooled on ice, and used immediately for qPCR. The reaction was carried out using an ABI QuantStudio 6 Flex fluorescent PCR instrument with an SYBR Premix EX Taq kit. The experiment used 10 μL of 2× qPCR Mix, 0.5 μL of each primer, 0.5 μL of cDNA, and 8.5 μL of ddH_2_O. The amplification program was set as follows: 95 °C 300 s followed by 40 cycles of 95 °C for 10 s and 60 °C for 30 s. The relative expression was calculated using the 2–ΔΔCT method [81], where T1 was used as a control to normalize the experimental data (T1, T2 and T3 represent 30DAF, 50DAF and 70DAF, respectively). Gene names and primer sequences are presented in Appendix A.

### 4.10. Statistical Analysis

GO and KEGG annotation of *A. trifoliata* using Eggnog (http://eggnog-mapper.embl.de/ accessed on 16 October 2022). Transcription factor identification of genome-wide proteins using Planttfdb (http://planttfdb.gao-lab.org/prediction.php accessed on 16 October 2022) [82,83]. Transcriptome and metabolome correlations were calculated using Python scripts and correlations were calculated using Pearson correlation coefficients (PCCs) [84]. Cytoscape v3.7.1 was used for co-expression network graphs for visualization [85]. Graphpad prism 6.0 (GraphPad Inc., La Jolla, CA, USA) was used for t-test analysis of metabolite content and visualization of metabolic content.

## 5. Conclusions

In this study, we identified the NAC family from *A. trifoliata* genome. One hundred one AktNACs were divided into eight subgroups. These NAC genes were unevenly distributed on 16 chromosomes. Genes with similar conserved motifs and gene structures clustered phylogenetically into one group. cis-acting elements of the AktNACs gene promoters were associated with plant hormones, organ development and abiotic stresses. Transcriptome analysis showed that *AktNAC060*, *AktNAC085*, *AktNAC071* and *AktNAC095* showed upregulated at all three stages. Combined transcriptome and metabolome analyses indicated that AktNACs synergistically regulate the synthesis of seed metabolites with a variety of transcription factors. These candidate genes provide an important theoretical basis for future molecular breeding of AktNACs. These candidate genes provide an important theoretical foundation for future molecular breeding of AktNACs.

## Figures and Tables

**Figure 1 plants-12-01518-f001:**
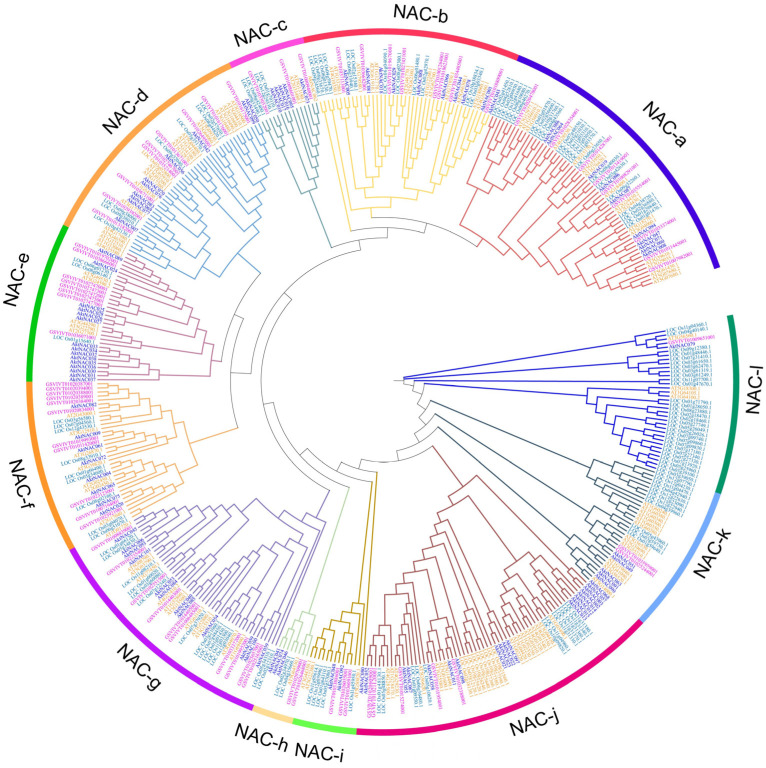
Phylogenetic tree analysis of NAC protein in *A. trifoliata* (101), grape (71), rice (141), and *Arabidopsis thaliana* (115). We used the neighbour-joining method to construct the phylogenetic tree, with parameters set to the P-distance, pairwise deletion, and bootstrap value of 1000 replicates, and other parameters by default. The blue, purple, orange, and bottle green represent the NAC proteins of *A. trifoliata*, *Arabidopsis thaliana*, grape, and rice respectively.

**Figure 2 plants-12-01518-f002:**
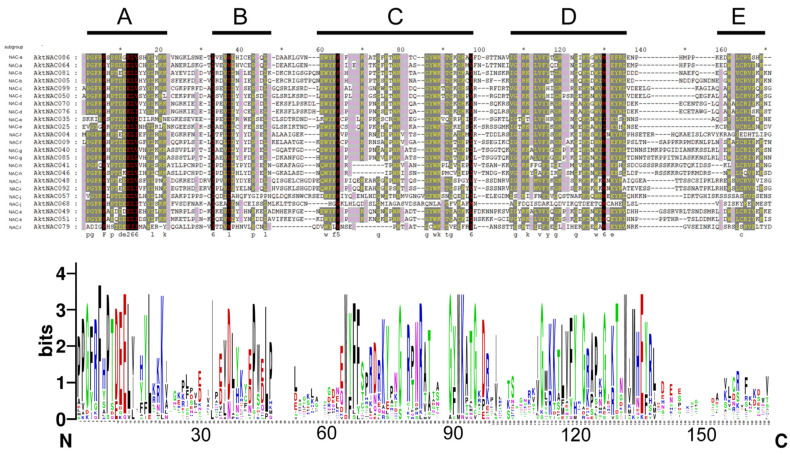
The conservation of AktNAC amino acid residues was analyzed. The depth of color is used to represent conservation. A darker color indicates a higher conservation of amino acids or nucleotides at that position in the aligned sequences. A, B, C, D, and E are the five subdomains of the NAC domain. The *x*-axis represents the positions of amino acids, while the *y*-axis indicates the frequency of each amino acid at a specific position appears in a particular position. Different colors are used to represent different types of amino acids. “N” and “C” typically represent the amino terminus (N-terminus) and carboxyl terminus (C-terminus), respectively.

**Figure 3 plants-12-01518-f003:**
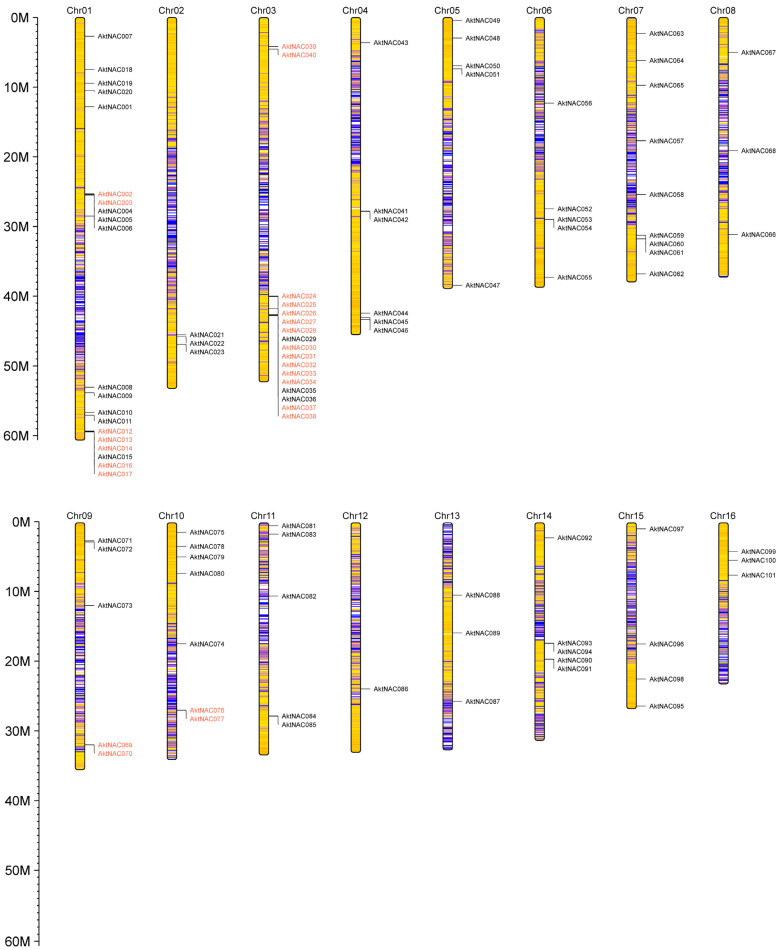
The chromosome locations of AktNACs genes. The chromosome numbers were indicated at the top of each chromosome. Red indicates tandem-duplicated genes. The location was drafted using Perl 5, version 18, subversion 4 (v5.18.4).

**Figure 4 plants-12-01518-f004:**
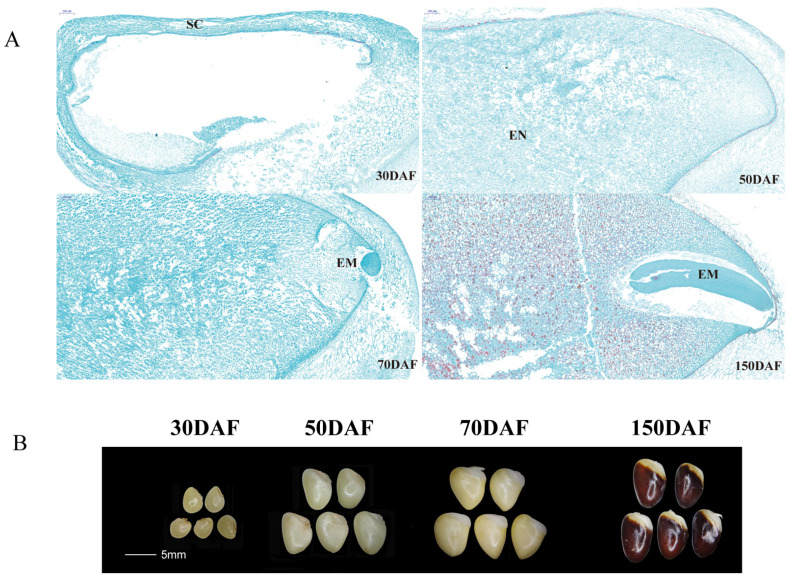
The seed development process of *A. trifoliata*. (**A**) Seed microstructure was observed under a laser confocal microscope. Scale bars: 200 µm. SC: Seed coat, EM: Embryo, EN: endosperm. (**B**) Seed development morphological changes was observed by SMZ800 stereomicroscope. Scale bars: 5 mm. 30DAF, 50DAF, 70DAF, and 150DAF represent 30 days, 50 days, 70 days, and 150 days after flowering, respectively.

**Figure 5 plants-12-01518-f005:**
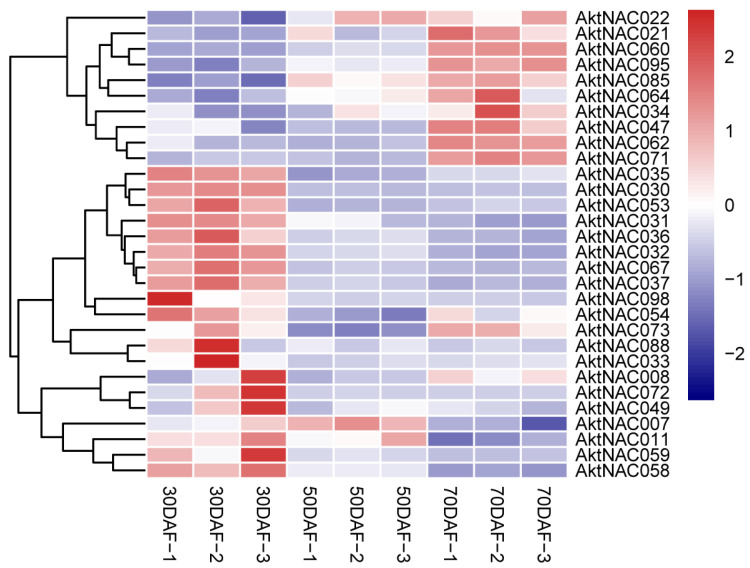
The expression levels of AktNACs in early seeds at different developmental stages. Each column in the figure represents a sample and each row represents a gene, normalized by gene expression FPKM values for the rows using the z-score method.

**Figure 6 plants-12-01518-f006:**
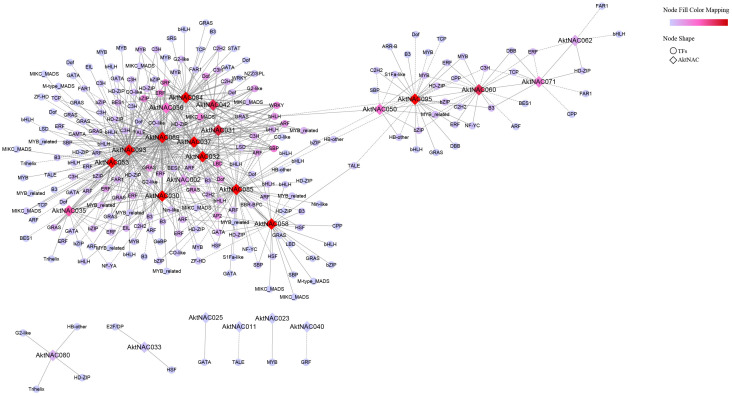
Co-expression network and enrichment analysis of AktNACs during early seed development of *A. trifoliata*. Co-expression network of AktNAC genes constructed based on transcriptome data during early seed development. Diamonds indicates AktNAC genes, round indicated transcription factors, and color shade indicates the number of associated genes.

**Figure 7 plants-12-01518-f007:**
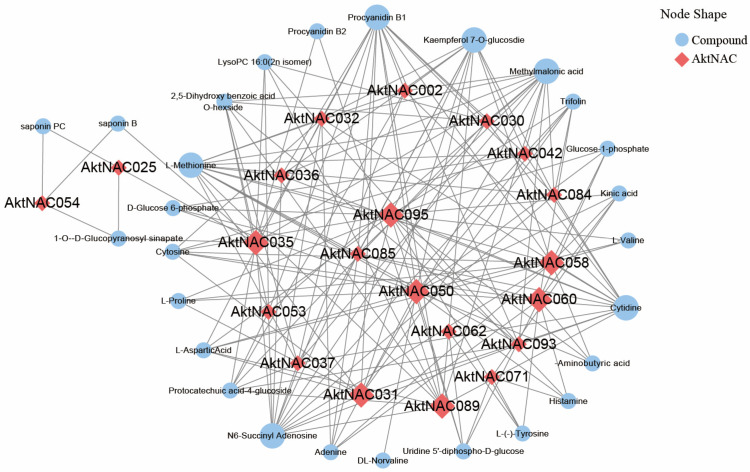
Co-expression network analysis of AktNACs and metabolites. Different shapes and colors represent different substances.

**Figure 8 plants-12-01518-f008:**
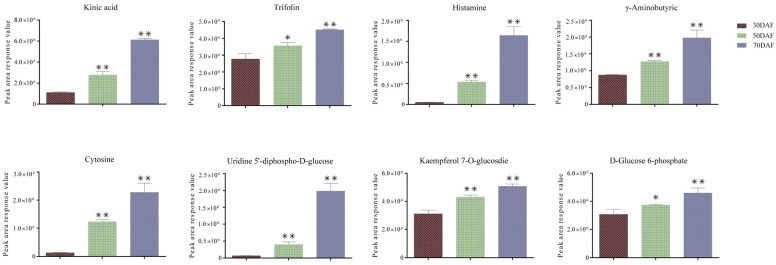
Metabolite content in the co-expression network of AktNACs and metabolites. Error bars denote the standard deviation of three replicates (* *p* < 0.05, ** *p* < 0.01).

**Figure 9 plants-12-01518-f009:**
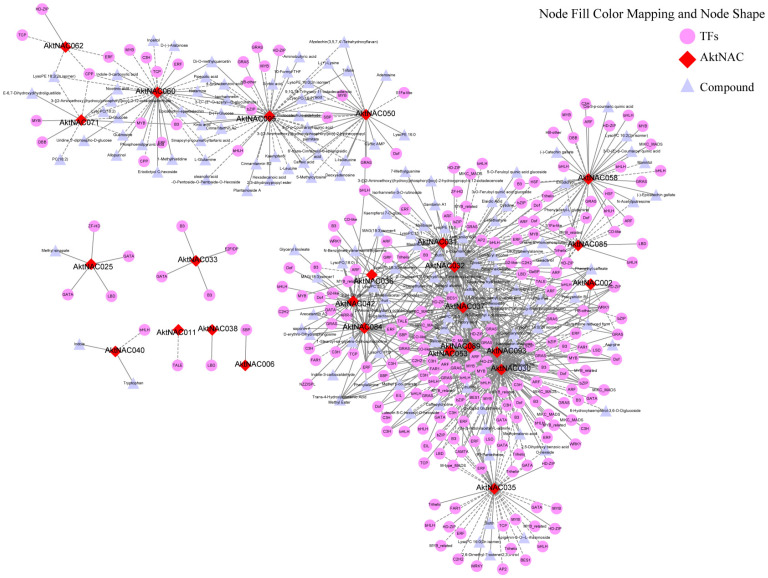
Co-expression of AktNACs, TFs and compounds.

**Figure 10 plants-12-01518-f010:**
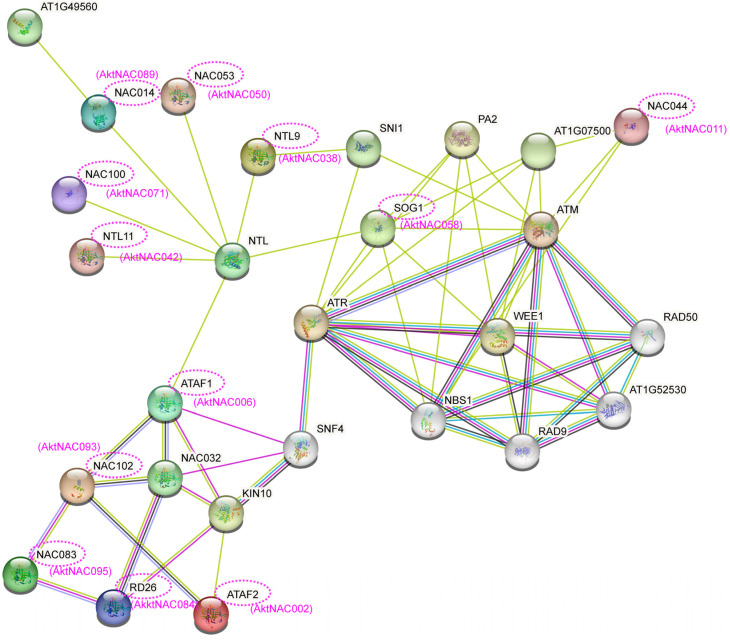
The interaction network of *Arabidopsis thaliana* NAC homologs proteins. The purple dashed oval indicated the *Arabidopsis thaliana* homologous genes to the *A. trifoliata* NAC protein, and the AktNACs were shown in purple below the *Arabidopsis thaliana* protein.

**Table 1 plants-12-01518-t001:** Correlation analysis of metabolites associated with co-expression of AktNAC095.

Gene Name	Compound	Molecular Formula	PCC
*AktNAC095*	Kinic acid	C_7_H_12_O_6_	0.982401 **
*AktNAC095*	Trifolin	C_21_H_20_O_11_	0.980908 **
*AktNAC095*	Histamine	C_5_H_9_N_3_	0.980678 **
*AktNAC095*	γ-Aminobutyric	C_4_H_9_NO_2_	0.973989 **
*AktNAC095*	Cytosine	C_4_H_5_N_3_O	0.969859 **
*AktNAC095*	Uridine 5′-diphospho-d-glucose	C_15_H_24_N_2_O_17_P_2_	0.962642 **
*AktNAC095*	Kaempferol 7-O-glucoside	C_21_H_20_O_11_	0.953569 **
*AktNAC095*	d-Glucose 6-phosphate	C_6_H_13_O_9_P	0.951671 **

** Means *p*-value < 0.01.

**Table 2 plants-12-01518-t002:** Correlation analysis of AkNACs co-expression network-regulated metabolites.

Node1	Node2	Correlation	Node3	Correlation
*mikado.Chr05G296* (B3)	*AktNAC060*	0.994325284 **	Kinic acid	0.990795 **
*mikado.Chr04G2027* (MYB)	*AktNAC060*	0.984943879 **	Uridine 5′-diphospho-d-glucose	0.984606 **
*mikado.Chr04G2027* (MYB)	*AktNAC060*	0.984943879 **	Histamine	0.983068 **
*mikado.Chr05G50* (GRAS)	*AktNAC095*	0.982596163 **	Kinic acid	0.982401 **
*mikado.Chr01G2874* (HD-ZIP)	*AktNAC095*	0.981437254 **	Trifolin	0.980908 **
*mikado.Chr01G2874* (HD-ZIP)	*AktNAC095*	0.981437254 **	Histamine	0.980678 **

** Means *p*-value < 0.01, Correlation > 0.9: Positive.

## Data Availability

The datasets of RNA-seq presented in this study can be found in online repositories (NCBI SRA with BioProject ID: PRJNA884501).

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
