# Peer review of "Identification of the NAC Transcription Factor Family during Early Seed Development in Akebia trifoliata (Thunb.) Koidz"

_plants, 2023, doi:10.3390/plants12071518_

Round 1

Reviewer 1 Report

In this manuscript, the authors focus their attention to a plant specific NAC transcription factors identified in Akebia trifoliate (since the group determined its genomic sequence). This is a large family in all plant species consisting of 101 members in Akebia, and on the structural base they have been divided further into 11 subfamilies. Furthermore, they analysed them in nearly every possible ways including to determine their chromosomal positions, gene duplication events, showing putative regulatory motif in their promoter regions, tissue specific and stress related expressions, as well as co-expression with other transcription factors and metabolites. They concluded that these AktNACs most probably have important regulatory roles in nearly all developmental processes including seed development.

Major comments

Obviously, the authors took the analyses of AktNAC factors very seriously, but my major concern is that as a reader it was not easy to follow the “story line” and it is rather look like pouring of too much information.

For example, Figure 2 is too complex (like Fig4,6,7-8) and hardly anything could see or read. I know that these are this type of analyses but I expect some clear outcome. I would rather suggest to show this figure in the supplementary, and do an analyses with selected members from each group to show the structural differences (you can see an example in Ooka et al., 2003 in Figure 4). I also would show the general structure of NAC factor in a figure.

Is there any further difference between these 11 subfamilies (like the level of expression and co-expression profiles)? I would rather try to structure the manuscript in this way if it is possible.

Alternatively, in comparison to Arabidopsis NACs what the authors could see and say? How AktNAC similar to Arabidopsis NACs and furthermore their structural and expressional similarities and differences? what about their subfamilies?  Some NAC TFs have already been characterized in Arabidopsis, and there are some well-known and functionally characterised members. SOG1 for example, which is suggested to be the plant p53 involved in DNA-damage response (Ascencio-Ibanez et al., 2009). Additionally, other related NAC factors like NAC044 was also shown to participate in DNA damage response, and recently it was shown to participate in multiprotein complexes consisting of other transcription factors (Lang et al., 2021).

Reviewer 2 Report

This manuscript investigated the possible function of NACs TFs by RNA-seq analysis during the A. trifoliata seed development. The whole work was simple, but it was quite interesting and meaningful. Thus, I think it could be accepted after several minor issues improved.

1. In title, point the particular metabolites or deleted it?

2. In Abstract, the possible function not the actual function, add ‘possible’ before ‘function’.

3. Line 27, the results in this work could not support the regulation statement, but only the relationship between NACs and metabolites, the authors should revise it.

4. Line 55-56, Solanum Lycopersicum, SlNAC1 and SlNAM1 should be written in Italic. Same mistakes in Line 108, Line 114, Line 120, Line 165-170, A. trifoliata. Line 221-231, gene names. Line 298, Line 316, the descriptions of ‘r’ and ‘p’, the figure caption of Figure 9. Please check the whole manuscript.

5. The important roles of these components such as organic acids, alkaloids, flavonoids, lignans coumarins, and terpenoid lipids during seed development should be introduced in somewhere.

Round 2

Reviewer 1 Report

 I recommend to publish the revised version.